# What Are the Characteristics of Torture Victims in Recently Arrived Refugees? A Cross-Sectional Study of Newly Arrived Refugees in Aarhus, Denmark

**DOI:** 10.3390/ijerph20146331

**Published:** 2023-07-10

**Authors:** Mette Hvidegaard, Kamilla Lanng, Karin Meyer, Christian Wejse, Anne Mette Fløe Hvass

**Affiliations:** 1Center for Global Health, Department of Public Health, Aarhus University, 8000 Aarhus, Denmark; kamillalanng@gmail.com (K.L.); wejse@clin.au.dk (C.W.); annhvass@rm.dk (A.M.F.H.); 2Department of Medical Diseases, Gødstrup Regional Hospital, 7400 Herning, Denmark; 3DIGNITY—Danish Institute Against Torture, 2100 København, Denmark; kamr@dignity.dk; 4Department of Infectious Diseases, Aarhus University Hospital, 8200 Aarhus, Denmark; 5Department of Social Medicine, Aarhus Municipality, 8000 Aarhus, Denmark

**Keywords:** torture, refugee, PTSD, health screening, refugee health

## Abstract

Torture victims live with complex health conditions. It is essential for the rehabilitation of torture survivors that their traumas are recognized at an early stage. The aim of this study was to investigate (i) the prevalence of reported torture exposure, (ii) the association between demographic characteristics and exposure to torture, and (iii) the association between PTSD and exposure to torture among recently arrived refugees in Aarhus, Denmark. Data were extracted from health assessments of refugees arriving in Aarhus in the years 2017–2019, and 208 cases were included in the analysis. The prevalence of reported torture was 13.9% (29/208). Most torture victims were found among refugees arriving from Iran (17.0% (9/53)), Syria (9.3% (8/86)), and Afghanistan (25.0% (5/20)). Significant associations were found between reported torture exposure and male gender, Southeast Asian origin, and a diagnosis of PTSD. In the study, 24.5% (24/98) of males and 4.5% (5/110) of females had been subjected to torture. However, it is possible that the prevalence of female torture survivors is underestimated due to the taboos surrounding sexual assaults and fear of stigmatization. Nearly half of the torture victims in the study were diagnosed with PTSD (44.8% (13/29)). The results confirm that torture victims constitute a vulnerable group living with severe consequences, including mental illness such as PTSD. Furthermore, understanding the cultural perspectives of the distress among refugees is crucial in providing appropriate healthcare services. This study highlights the importance of addressing the mental health needs of torture survivors and tailoring interventions toward vulnerable refugee populations.

## 1. Introduction

Despite governments worldwide agreeing that torture is a violation of human rights, torture tends to resurface when circumstances allow it. Torture can gain strength, become a culture, and be used as a weapon, terrorizing and controlling selected populations. Even in high-income countries, torture occurs, bearing witness to cracks in democracies. The acceptance of the use of torture on prisoners suspected of terrorism grew when it was presented as a necessity in “the war on terror” in the United States after the 9/11 attacks [1]. Furthermore, torture often strikes refugees while they flee to safety [2]. Although the United Nations (UN) in 1984 declared torture an inhumane act that should not be allowed under any circumstances [3], an Amnesty International report from 2015 [1] showed that torture or ill-treatment had occurred in 122 countries. Torture not only damages societies where it takes place; it is also present in countries receiving refugees. While torture is immediately harmful, victims are also forced to live with long-term consequences that may compromise their health and quality of life both physically [4,5,6] and mentally [7,8,9,10,11,12,13,14]. Many victims of torture live with chronic pain [5,6], and the risk of medical conditions such as diabetes and hypertension is increased [4]. Posttraumatic stress disorder (PTSD) and depression are common psychiatric diagnoses in those who have been tortured [7,8,10,12,14]. Family members of torture victims also suffer from trauma as secondary victims of torture [15,16].

Thus, the healthcare systems of the receiving countries face great challenges in treating torture victims with complex health conditions. It is essential for the rehabilitation of torture survivors that their traumas are recognized [17]. Recognition of the trauma history enables doctors to provide better healthcare for the refugees, including referring torture victims for further diagnostics and treatment. For instance, early intervention for PTSD could result in better outcomes [14]. However, a recent study [18] reported that only 55% of torture survivors in Denmark had been asked by their general practitioner if they had been exposed to torture. A study from 2008 found a prevalence of torture victims of 45% among newly arrived refugees in Denmark [19], whereas a 2020 study by Andersen et al. found a prevalence of 27% [20]. Updated research on the prevalence of torture victims is scarce. To effectively help survivors to be referred to relevant departments in the healthcare system and thereby facilitate rehabilitation of physical and mental injuries inflicted by torture, it is essential to identify the torture survivors upon arrival [17,21]. This raises the question of whether certain characteristics are present among tortured refugees arriving in Denmark. Furthermore, since torture victims have previously been described as a group with a high risk of developing PTSD [14], this might affect some of the newly arrived refugees.

To address these matters, this cross-sectional study aimed to investigate the (i) prevalence of reported torture exposure, (ii) association between the demographic characteristics of country of origin, as well as gender and age and exposure to torture, and (iii) association between PTSD and exposure to torture among recently arrived refugees in Aarhus, Denmark based on health assessments conducted in the AARHAUS study (Aarhus Refugee Health Assessments Using Systematic approaches). We conducted a cross-sectional study based on information from general health assessments of newly arrived refugees in Aarhus.

## 2. Materials and Methods

The Danish Data Handling Authority authorized the project (file number 2015-55-0586), as well as the Danish Patient Safety Authority (file number 3-3013-2624/1). The Central Denmark Region Committee on Health Ethics assessed the project and determined that approval was not required.

### 2.1. Participants

Three groups of refugees typically arrive in the Municipality of Aarhus: (i) quota refugees through the UNHCR, (ii) refugees with asylum-seeking backgrounds, and (iii) refugees arriving through a family reunification system. The term “refugee” used in this paper refers to all three groups. Quota refugees are refugees identified by the UNHCR as having urgency of resettlement based on protection needs, vulnerabilities, and special needs [22]. The identification of quota refugees is a method to convey the urgent need of resettlement of refugees from the country to which they have first fled, where their fundamental rights are in danger. Through the years 2017–2019, the most common nationalities of asylum applicants in Denmark were Syrian, Eritrean, Iranian, Moroccan, Afghan, Georgian, and stateless [23]. 

Refugees arriving in Denmark are distributed randomly across municipalities. Having been granted permanent residence status, all refugees in the Municipality of Aarhus are offered a voluntary general health assessment performed by doctors in the municipality [24]. Close to 90% of the arriving refugees accept the offer of a general health assessment [25]. Refugees included in this study arrived in Aarhus in 2017, 2018, and 2019, and are therefore considered a representative sample of all refugees arriving in Denmark in this timeframe.

Only refugees at or above 18 years of age at the time of the general health assessment appointment were included. Refugees who missed the appointment for the general health assessment were excluded. Only the cases with available information on exposure to torture were included in the statistical analysis.

### 2.2. Procedures

The general health assessments consist of a structured interview about background, physical and mental health, a physical examination, and a comprehensive blood sample panel. The assessments take place at the medical clinic in the Municipality of Aarhus and are carried out by doctors working at the municipality. The interviews are conducted with help from interpreters, using the native language of the refugee.

The interview includes screening questions about exposure to trauma, such as torture before or during flight. The interview also includes screening questions for PTSD, including symptoms of arousal, avoidance, intrusion, and negative alterations in cognitive mood, as well as traumatic events before or during fleeing [26]. If the refugees are suspected of PTSD based on the screening questions, and if consent is given by the refugee, they are referred directly to a specialized PTSD clinic at the Department of Psychiatry (DOP), Aarhus University Hospital. Here, trained psychiatrists conduct formal diagnostics (using ICD-10 standards) and treatment of refugees with PTSD. The specialized PTSD clinic also treats torture victims. If other psychiatric diagnoses are suspected, the refugee’s general practitioner is informed and advised to refer them for formal psychiatric diagnostics at a specialized psychiatric department [26].

We received data from DOP on all refugees with referral and formal diagnosis of PTSD. General health assessments performed on refugees in Aarhus from 1 January 2017 to 1 January 2020 were reviewed.

### 2.3. Measures

For research purposes, REDCap (Vanderbilt, USA) was used to establish a database with demographic information, including gender, age, and country of origin. The identification of torture victims included in the statistical analysis in this study took place retrospectively based on the information acquired in the general health assessments. To ensure consistency with the international definition of torture in the UN declaration from 1984 [3], a standard tool to identify torture victims developed by DIGNITY (Danish Institute against Torture) and The Danish Red Cross [27], was used to identify torture survivors when reviewing the general health assessments. The criteria defined in this tool were added to the database. The criteria regarding torture are shown in Table 1, and to each query, the answer categories available were “Yes,” “No,” or “No data available.” First, the following three variables were addressed. “Arrested, retained or imprisoned?” “Severe violence, threats or degrading treatment?” “Witnessed severe violence or degrading treatment against others?”

Next, narrative criteria were considered. (1) Subjected to severe pain or suffering, physically or mentally? (2) Was it done intentionally? (3) Was there a purpose to the action? (4) Was the action performed by a representative of the authorities or “those in power”?

Finally, “Torture exposure?” (if “Yes” to parameters 1–4) was concluded.

Three people were responsible for extracting data on demographics and torture from the general health assessments. Cases were discussed when in doubt about torture exposure. Some cases were discussed with clinicians from DIGNITY, aiming to ensure compliance with the UN definition of torture [3].

Exposure to torture in this paper includes both torture before migration and during flight. When the database was established, information on the location and timeline of torture was not included. It is therefore not possible to distinguish between torture experienced in the country of origin or during flight in the statistical analysis.

Information on referral to the PTSD clinic and information from the PTSD clinic regarding diagnosis of PTSD were added to the database.

The content of the interview used in the General Health Assessment is based on a ministerial guideline on assessment of newly arrived refugees [28]. The guideline highlights how refugees are often strangers to the Danish healthcare system and due to their cultural backgrounds have a different framework for understanding the healthcare context. For this reason, it is recommended that doctors begin conversations with information about the rules of confidentiality. The guideline includes recommendations on obtaining a thorough medical history with questions about experiences before and during flight, including spending time in refugee camps, violence, abuse, trauma, and torture exposure. Further, it contains questions covering PTSD symptoms and other consequences of trauma.

### 2.4. Data Analysis

Using Pearson’s chi squared test, a bivariate analysis was performed to determine whether reported exposure to torture and country of origin, gender, and age were statistically significantly related, with *p*-values below 0.05 considered statistically significant. For smaller populations, Fisher’s exact test was used. Microsoft Excel (2018) was used for all calculations. The number of associations controlled for in the statistical analysis was not >20, and consequently we refrained from controlling for multiple comparisons with the Bonferroni correction.

## 3. Results

### 3.1. Participants’ Characteristics

From 1 January 2017 until 1 January 2020, 470 refugees participated in a general health assessment in Aarhus. A total of 214 refugees were below the age of 18 years and thus excluded from the study. The total study population consisted of 256 adult refugees. It was possible to answer the question of torture in 208 cases. In 48 cases, it was not possible to conclude whether the refugee had been exposed to torture or not due to missing information. Data on age and gender were available for all the included cases. For one included case, the country of origin was not available.

Table 2 contains demographic data for the study population, including countries and UN regions of origin [29]. The overall median age and gender distribution in the groups with and without information on torture was similar to the total study population. Therefore, the group where information about torture exposure was available (*n* = 208) is considered representative of the total study population (*n* = 256).

The study population originated from six different UN regions and fifteen different countries (Table 2); one refugee was of unknown origin. The biggest group of refugees came from West Asia (49.0% (102/208)), with the majority originating from Syria (41.3% (86/208)). The second-largest group came from South Asia (35.1% (73/208)), mainly from Iran (25.5% (53/208)). Smaller groups of refugees originated from East Africa, North Africa, Southeast Asia, and Eastern Europe.

### 3.2. Descriptive Analysis

Of the 208 cases with information on torture exposure, 179 refugees reported they had not been exposed to torture, while 29 refugees had been exposed (Table 2). This equals a prevalence of 13.9% (29/208) of refugees subjected to torture.

When considering the prevalence of torture victims in each region, Southeast Asia was the region with the highest percentage (100% (2/2)), as both of the two cases in this group had been subjected to torture. South Asia had a high prevalence of torture exposure as well (19.2% (14/73)), with a prevalence of 25% (5/20) among Afghan refugees and 17% (9/53) among Iranian refugees. A lower prevalence was detected among refugees originating from West Asia (9.8% (10/102)) and East Africa (12% (3/25)), while none of the refugees originating from North Africa or Eastern Europe were victims of torture. The refugee of unknown origin had not been subjected to torture.

Probability values from bivariate analyses comparing country and region of origin and reported exposure to torture are shown in Table 2. A statistically significant result was found (chi^2^ *p*-value < 0.000) for refugees originating from Southeast Asia.

The gender distribution in the group of refugees with information on torture was close to being equal, with women constituting 52.9% (110/208) (Table 2). However, among the reported tortured refugees, 82.8% (24/29) were male. Among the male cases, 24.5% (24/98) had been subjected to torture, while the number was 4.5% (5/110) among the female cases. When analyzing the relation between gender and reported exposure to torture, a statistically significant result was found (chi^2^ *p*-value < 0.000) for male gender.

Figure 1 includes participants where information on torture was available. It depicts the proportion of males and females reported to have been exposed to torture from each region of origin. While women account for the majority of the refugees, proportionally few are reported victims of torture. Meanwhile, men are overrepresented in cases exposed to torture.

The study population was divided into five age groups: 18–25 years, 26–35 years, 36–45 years, 46–55 years, and ≥56 years. The age group from 26–35 years was by far the biggest (35.1% (73/208)), while the rest of the refugees were distributed as follows: 18–25 years (21.6% (45/208)), 36–45 years (15.9% (33/208)), 46–55 years (18.8% (39/208) and ≥56 years (8.7% (18/208)). The highest percentage of reported torture was in the age group 26–35 years (17.8% (13/73)). The lowest percentage of reported torture was found in the youngest age group, from 18–25 years (6.7% (3/45)). When comparing age groups, both separately and combined, to torture exposure, no statistically significant results were found in the bivariate analyses.

Of the reported tortured refugees, 44.8% (13/29) were diagnosed with PTSD. Table 3 shows results from the statistical analyses on health outcomes. Statistically significant results were found in the bivariate analyses, both between reported torture exposure and referral to the PTSD clinic (chi^2^ *p*-value < 0.000) and between reported torture exposure and diagnosis of PTSD (chi^2^ *p*-value < 0.000) established at the clinic. No statistically significant associations were found between adjustment disorders (defined as ICD-10 F43.2–43.9) or major mood disorders (defined as ICD10 F30–F39) and reported torture exposure. A statistically significant association was found between anemia and not being exposed to torture. We suspect that this is a random finding. No further associations were found when comparing health outcomes and torture exposure.

Further information on the specific variables is available in other publications from the AARHUS cohort [24].

## 4. Discussion

This study aimed to investigate the (i) prevalence of reported torture exposure, (ii) association between the demographic characteristics of country of origin, gender, and age and exposure to reported torture, and (iii) association between PTSD and exposure to torture among recently arrived refugees in Aarhus, Denmark. The statistical analysis included data from 208 general health assessments performed on adult refugees in Aarhus in 2017–2019.

A total of 13.9% (29/208) of the study population reported they had been subjected to torture. Previous studies have found the overall prevalence of tortured refugees and immigrants arriving in Denmark to be between 5% and 35% [18]. In 2008, the prevalence of tortured newly arrived refugees in Denmark was 45%, according to the medical group from Amnesty International in Copenhagen [19]; in 2020, the prevalence reported by Andersen et al. 2020 [20] was 27%. The study by Amnesty International [19] is older and the circumstances for refugees different from today, and in contrast to the present study, there were refugees from Chechnya in the study population. Torture might have been more common in Chechnya compared to the countries of origin in this study. Further, Amnesty International [19] used a method more directly focused on acquiring information on torture, whereas this study and the recent study by Andersen et al. [20] collected data retrospectively from general health assessments, where questions on torture exposure were occasionally part of a screening, which could have resulted in lost information on torture and thus underestimation of the prevalence of torture. Conversely, in the study by Andersen et al. [20], the prevalence of torture was closer to the prevalence in this study, possibly reflecting a more accurate status for refugees arriving in Denmark today.

This study included immigrants arriving through family reunification for refugees, which may also explain the difference in the prevalence of torture victims. Finally, the present study included a larger study population than both of the earlier studies, which could also explain a lower prevalence. In a systematic review from 2016 by Sigvardsdotter et al. [30] reviewing international literature, the prevalence of reported torture ranged between 1% and 76%. However, almost 14% of arriving refugees have a history of reported torture, which should call attention to the subject.

Even with the slightly higher participation of women (52.9% (110/208)) in this study, 82.8% (24/29) of the reported tortured refugees were men. A significant association was found between being male and being a victim of torture (chi^2^ *p*-value < 0.000). This is consistent with numerous previous findings of men being more likely to be subjected to torture [14,18,31,32]. A recent larger study by Ostergaard et al. [18] found that 70% of tortured victims were male. An extensive review from 2019 [14] also found that men more often had been exposed to torture compared to women; only a few studies found the opposite. Meanwhile, women might be more likely to be asked about torture history [18]. Although both men and women risk sexual torture, women face particularly comprehensive consequences from this type of abuse. It should further be noted that women have been found numerous times to have a higher occurrence of sexual torture exposure [14,33,34,35,36]. As this type of torture is likely to be kept secret [34,37] due to feelings of shame, guilt, and fear of stigmatization, it is possible that the prevalence of female torture survivors is underestimated. Moreover, the consequences of sexual abuse entail an impact on the perception of the abused woman by society with potential pregnancy, possible inability to have children, or loss of virginity [38]. According to a publication by DIGNITY from 2015 [39], detained women are (compared to men) at greater risk of gendered torture, including rape, sexual abuse, and violence against pregnant women. Prison regimes are usually aimed at male inmates and guarded by male prison staff [38]. In addition to the risk of sexual torture, women are exposed to sexual humiliation, including unnecessary body searches or being watched during intimate moments by male guards [38]. In addition, sexual torture might not leave physical injuries bearing witness to the traumatic event, as with other types of torture [40], which may also lead to an underestimation of female torture survivors. Further, as witnessing torture on others also falls within the definition of torture, this may cause further underestimation, as it was not systematically assessed during the health assessments.

The highest numbers of torture victims were found in refugees from Iran, Syria, and Afghanistan. Common to these nations is the known utilization of torture by the government to control and repress those who criticize authorities and other civilians [41,42,43]. Characteristic of Syria and Afghanistan is also the use of torture by non-government groups [41,42]. When the conflict arose in Syria in 2011, peaceful demonstrations were answered with violent actions by government forces, and the occurrence of torture drastically increased [42]. Tens of thousands of Syrians have been victims of enforced disappearance, and of these, the majority have been civilians [44]. Torture, violence, and rape are used routinely during detention to terrorize captives. Most commonly, the torture is carried out by Syria’s four intelligence services: Air Force Intelligence, Military Intelligence, Political Security, and General Intelligence (also referred to as State Security), as well as the Military Police [42]. The UN Commission of Inquiry has reported that torture has also been carried out by non-state armed groups such as IS and Jabhat al-Nusra [42].

The political and social climate in Afghanistan has been influenced by the weak implementation of the judicial branch in the formally declared democracy and by the armed conflict between the state and insurgent groups [41]. Civilians were risking exposure to torture both by institutions representing the government and anti-government groups, such as the Taliban and ISIS-K [41]. Some aspects of the social setting were the backdrop for violence and sexual abuse against women and children—especially boys. Boys were sexually abused by security forces in all provinces of Afghanistan [41], and sexual abuse of children by school personnel was also reported [41]. Social norms and non-compliance with law by officials made it difficult and dangerous for children, women [41], and human rights defenders [41] to report sexual abuse to authorities. Women committing social offenses by eloping from violent or oppressing social circumstances were often arrested and charged with “immorality” [41]. Other exposed groups were LGTBI individuals and members of religious and ethnic minorities, e.g., Shia Hazaras, Hindus, and Sikhs [41], who were discriminated against, harassed, and physically attacked. Hazaras were especially violently attacked and killed by ISIS-K [41].

The government of Iran is ultimately controlled by a ranking Shi’i clergy, and all legislation is to be passed by the 12-member Council of Guardians that validates compliance with the standards of Islamic law [45]. According to the 2016/2017 report on Iran by Amnesty International [43], torture was commonly used by the Iranian Ministry of Intelligence and the Revolutionary Guards during detention and interrogation. The Iranian authorities were discriminating against religious and ethnic minorities in law and instigated and supported persecutions. Members of the Baha’i religious minority and ethnic minorities such as Kurds, Ahwazi Arabs, and Azerbaijani Turks were arbitrarily arrested [43]. In particular, detained ethnic minorities were subjected to torture, ill-treatment, and grossly unfair trials. Detainees were commonly tortured primarily in order to coerce confessions that were widely admitted as genuine evidence in courts. Further, prolonged solitary confinement was routinely applied [43]. Women and girls were discriminated against in law, e.g., insufficient protection against sexual violence, domestic violence, and forced and early marriages. Another example is the compulsory veiling laws that allowed police and paramilitary forces to subject women to violent harassment and arrests [43]. Punishments amounting to torture—especially flogging—were used as punishment for acts such as mixed-gender partying and protests against employment conditions and also as punishment for crimes of violence and robbery. In the latter case, punishments also included blinding and amputations [43]. In general, trials were unfair to defendants, e.g., hindering their access to lawyers and family, denying lawyers access to case files, and allowing confessions extracted under torture as evidence in trials [43].

A high number of refugees in this study arrived from Eritrea. A result of the political processes in Eritrea since independence from Ethiopia in 1993 is the election of a president by the first interim legislative body. The president, who is still residing, is head of the only political party in Eritrea [46,47]. A constitution including a chapter on fundamental rights was adopted in 1997, but was never enforced, and the president announced the preparation of a new constitution which has a status not clarified by Eritrean authorities [43]. In Eritrea, only four religions—Islam and the Orthodox, Catholic, and Lutheran churches—are recognized. Eighteen months of military duty is compulsory, but conscripts are never demobilized, and conscript labor was used in mining and construction at low pay [43]. Citizens were arbitrarily arrested and detained because they tried to avoid conscription, e.g., by fleeing the country, practicing a religion not recognized by the state, or were perceived to have expressed political dissent [43]. Prisoners were tortured by Eritrean authorities for political and religious activities, attempts to flee the country, and for failing military service [43]. Ethnic minorities, especially Kunamas and Afars, were victims of discrimination by both the government and society, and members of the Kunama group were detained [47]. Moreover, forcibly returned asylum seekers were tortured and threatened to be tortured [48]. The 2017 Country Report on Eritrea by the United States Department of State [47] found high numbers of women in early marriages and young birth ages. Further, women and girls were to a large extent victims of sexual violence amounting to torture by military personnel in military camps and the army [47].

A significant association between refugees of Southeast Asian origin and reported exposure to torture was found. However, this finding is based on only two refugees. A bigger study population is necessary to confirm this association. In 2015, Amnesty International reported that torture continued to occur in every region of the world [1]. Previous Danish studies have investigated the origin of torture survivors, where high incidence was found among immigrants and refugees originating from the Middle East [35,49] and Asia [35,49,50]. Similarly, Ostergaard et al. [18] found an increased risk of torture among male immigrants originating from the Middle East and North Africa compared to Southeast Europe. However, the study found a decreased risk of torture for males originating from East Asia compared to males originating from Southeast Europe [18].

Of the torture victims, almost half were diagnosed with PTSD. Associations were found between being a victim of torture and being referred to the PTSD clinic and diagnosis of PTSD. These associations confirm previous findings of high PTSD rates in torture victims [8,14]. In a review by Shuhaiban et al. [14], which included 12 studies, most studies reported PTSD rates of 50% or higher among torture victims. A large meta-analysis by Steel et al., including 161 articles, found torture to be the strongest factor associated with PTSD [8]. Andersen et al. [20] also found an association between torture and signs of PTSD. Along with a traumatic event triggering PTSD among torture victims, it has been found that living conditions after seeking refuge also play a decisive role in the mental state of those suffering from PTSD [51]. Momartin et al. [51] found that temporary compared with permanent protection visas were associated with impaired health. Specifically, a temporary compared with a permanent protection visa made a substantial additional contribution to PTSD symptoms. In Denmark, in most cases, it is not possible for refugees to apply for a permanent residence permit until they have had a temporary residence permit for eight years (Danish Immigration Service, n.d.). This places refugees with PTSD with unclarified or a temporary residence permit in a vulnerable position with a likely chance of PTSD symptoms persisting or worsening.

The sample included in this paper is culturally heterogeneous. When assessing the health of refugees, distress can be expressed in idioms other than those most commonly known among healthcare staff in the Western world [52,53]. For instance, in Syria, from where the biggest group in this sample originated, “mental health” is not commonly understood as a concept [52]. Suffering is perceptualized as a part of life, and further impaired mental health is negatively associated. Somatic and psychological symptoms are interlinked. Hence, this may cause healthcare staff to overlook or misunderstand symptoms of distress and trauma history when assessing refugees.

Torture victims constitute a vulnerable group of patients for several reasons. Although no significant associations were found in this study, it is well established that mental health disorders besides PTSD, such as depression and anxiety, are common diagnoses among victims of torture [7,8,10,12,14]. Increased risk of suicidal behavior [9,11], neurological sequelae [13], and chronic pain [5,6] are other challenges faced by these victims. Torture also increases the risk of hypertension and diabetes [4]. Not only does torture severely affect the victim; the family and loved ones of the tortured also live with consequences. Secondary victims of torture have a higher risk of mental illness [16] and possibly of intimate partner violence perpetrated by the torture victim [15].

It is a crucial step towards effective healthcare to address the question of torture, as it could affect the physical and mental health of torture victims if unrecognized [17]. Furthermore, early intervention could be an important factor in the treatment of refugees, such as the treatment of PTSD [14]. This makes health screening and thorough healthcare for refugees soon after arrival in the recipient country critical. Other predictors of improved well-being of torture survivors are social support [54,55,56] and stable housing [57]. Detention [50,58] is associated with impaired health.

Although the torture itself took place far from the recipient country, the consequences trouble the arriving torture victims with complex conditions, affecting both their physical and mental health. This places the healthcare system in a unique position as it gives healthcare staff an opportunity and an obligation to help. To treat these patients who fled their country because of unstable life circumstances such as war and prosecution, other approaches may be necessary than when treating patients who have grown up in a peaceful country with a well-functioning healthcare system. When helping refugees, doctors and other healthcare staff should be prepared to address the matter of torture, since this could be fundamental in treating these patients and improving their future physical and mental health.

Denmark ratified the UN Convention against Torture [3] in 1987 and has therefore committed to the implementation of assessment and rehabilitation of torture victims. According to UN Comment No. 3 on the UN Convention against Torture [59], states are obligated to provide redress to torture victims. Redress entails restitution, compensation, rehabilitation, satisfaction, and guarantees of non-repetition [59]. Hence, recognizing someone as a torture victim opens up a range of rights and opportunities for the refugee. Additionally, status as a torture victim can be a decisive element in an application for a residence permit. It is therefore crucial that refugees are offered a comprehensive assessment of torture exposure, complying with the UN Convention against Torture [3].

Although Danish research on torture is abundant, recent data from newly arrived torture victims are scarce. The significant associations identified in this study between torture exposure and male gender, Southeast Asian origin, and a diagnosis of PTSD shed light on potential characteristics that may contribute to identifying torture victims. The findings reinforce the critical relationship between torture exposure and the development of PTSD. The implications of this study highlight the urgent need for policies and comprehensive rehabilitation programs that address the complex mental health conditions faced by torture victims. By recognizing the specific vulnerabilities of certain subgroups and tailoring support systems accordingly, policymakers, healthcare providers, and humanitarian organizations can work collaboratively to provide effective rehabilitation and care for this vulnerable population.

Further research on the prevalence of torture and associations between demographics and torture exposure is needed.

## 5. Limitations

A little more than 10% of the arriving refugees declined to participate in the general health assessment. It is possible that this group has the most severe health consequences, including those originating from torture. This may have resulted in an underestimation of torture exposure. However, it is also possible that refugees who refused did not feel the need for such an evaluation and thus were less likely to have been exposed to torture. Of the 256 refugees in the study population, it was unfortunately not possible to determine the question of torture in 48 cases. The lack of information in these cases may indicate that there was little physical or mental evidence to suggest the possibility that a refugee may have been a torture victim. This reduced the number of cases included in the statistical analysis to 208. If a larger study population had been included, the probability of finding further associations would have been bigger.

Data from the general health assessments were collected retrospectively. Therefore, the quality of the information on torture depends on how the general health assessments were carried out. Although the interview conducted in the general health assessment is based on ministerial guideline with recommendations, information was not collected using a consistent systematic protocol on all refugees. Likewise, even with the use of a ministerial guideline, the possibility of social desirability bias affecting the results of the statistical analysis is not eliminated with a sensitive topic such as torture exposure when relying on self-reported data. The general health assessments aim to establish a broad picture of the health of the refugee, focusing on many aspects other than torture. In some cases, the question of torture was not discussed with the refugee due to other topics being prioritized, lack of time, or because the refugee refused to discuss the subject. The structured interview was always performed by a trained clinician, and several clinicians conducted the general health assessments. Clinicians have individual ways of structuring and prioritizing interviews with refugee. In a few cases, the refugee arrived late for the interview, reducing the time for conversation. These factors may have resulted in differences in the quantity and quality of the information on torture.

Three people conducted the extraction of data from the general health assessments. Even though a standardized tool was used, the risk of differences in the interpretation of the general health assessments is not eliminated.

The refugees in this study originated from six different UN regions and fifteen different countries, representing a heterogeneous group. They came from cultures and traditions with views on and understanding of health and healthcare systems possibly very different from European and in particular Danish culture. It has previously been found that culture may influence the assessment of torture survivors [60,61], which may have resulted in missing information. Although all the refugees were informed at the beginning of the structured interview that the conversation was confidential, the trust in the healthcare system and authorities might be compromised for refugees fleeing their home country. This could be particularly present among refugees subjected to torture [34], which, by the UN’s definition, is performed by a public official or another person acting in an official capacity (Convention against Torture and Other Cruel, Inhuman or Degrading Treatment or Punishment, 1984). The clinical setting itself may act as a reminder of experienced torture, consequently preventing some refugees from wishing to have any contact with health professionals. Similarly, it is possible that some of the 48 with missing information on torture actually were torture victims who chose not to talk about their traumas. Furthermore, it is known that torture can cause psychological dissociation as a response to the traumatic event, resulting in difficulty accessing the memory [62,63]. Hence, many refugees report traumatic events much later, even though their asylum claim might benefit from reporting them upon arrival [63]. Finally, as partial or complete amnesia of a traumatic event is a main symptom of PTSD [64], some refugees might not have been able to share their experiences with torture, referred to a PTSD clinic, or diagnosed with PTSD, resulting in recall bias. Conversely, refugees referred to the PTSD clinic might have been more likely to provide detailed information on torture exposure than refugees who had been exposed but showed no symptoms at the time of the assessment.

This study was carried out using a cross-sectional approach, and hence it is not possible to conclude causal relationships based on the statistical analysis. Overall, 208 cases were included in the statistical analysis, which is a significant amount. However, with an even bigger sample, the risk of chance findings would be reduced even further.

The study was not controlled for multiple comparisons; thus, some associations may be statistically significant by chance.

## 6. Conclusions

This cross-sectional study found that a significant proportion of newly arrived refugees in Aarhus reported having been subjected to torture. Men constituted the highest proportion of those tortured. The results confirm previous findings on torture victims being vulnerable patients with many severe and complex health conditions, including PTSD. This poses a great challenge to healthcare systems in countries receiving refugees when diagnosing and treating torture victims.

## Figures and Tables

**Figure 1 ijerph-20-06331-f001:**
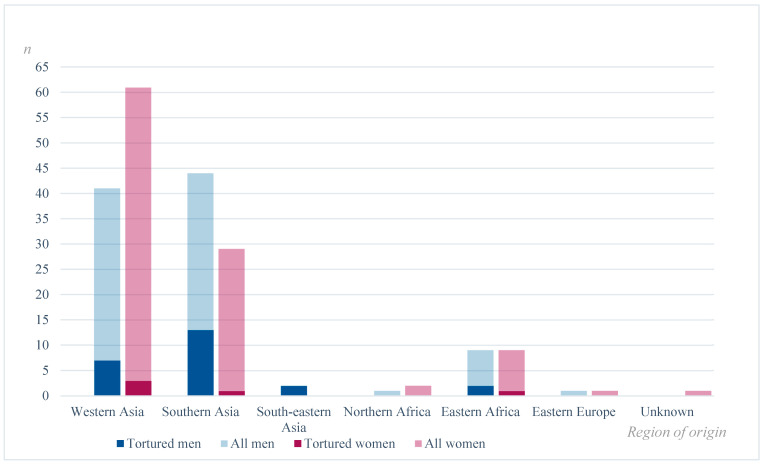
Numbers (*n*) of reported tortured men and women and men and women in total by region of origin.

**Table 1 ijerph-20-06331-t001:** Criteria of torture exposure. Based on Tool for identification of torture survivors by DIGNITY and Danish Red Cross).

Criteria of Torture Exposure
Arrested, retained or imprisoned?
Severe violence, threats or degrading treatment?
Witnessed severe violence or degrading treatment against others?
Narrative criteria
1. Subjected to severe pain or suffering, physically or mentally?
2. Was it done on purpose?
3. Was there a purpose of the action?
4. Was the action performed by a representative of the authorities or ‘those in power’?
Conclusion
Torture exposure? (yes = yes to 1–4)

**Table 2 ijerph-20-06331-t002:** UN regions and countries of origin according to the M49 standard (United Nations, 1999).

Region of Origin	Country of Origin	N (%)	Median Age (Range), Years	% Female	Reported Exposure to Torture, *n*(%) [95%CI]	Chi^2^ *p*-Value
Western Asia		102 (49.0%)	36 (18–70)	59.8	10 (9.8%) [4.2;15.8]	0.09098
	Iraq	8 (3.8%)	24.8 (21.8–32.2)	62.5	1 (12.5%) [−5.9;7.9]	1.0 *
	Syrian Arab Republic	86 (41.3%)	37.1 (18–70)	59.3	8 (9.3%) [2.3;13.7]	0.10480
	Other #	8 (3.8%)	49.2 (21.8–65.6)	62.5	1 (12.5%) [−5.9;7.9]	1.0 *
Southern Asia		73 (35.1%)	33.5 (18.2–63.7)	39.7	14 (19.2%) [6.0;22.0]	0.10893
	Afghanistan	20 (9.6%)	33.5 (19.8–61.4)	45.0	5 (25.0%) [−4.6;14.6]	0.167 *
	Iran (Islamic Republic of)	53 (25.5%)	33.5 (18.2–63.7)	37.7	9 (17.0%) [1.3;16.7]	0.45939
Southeast Asia		2 (1.0%)	30.4 (24.5–36.3)	0.0	2 (100%) [−17.4;21.4]	0.019 *
North Africa		3 (1.4%)	40.5 (37.8–48.4)	66.7	0 (0.0%)	1.0 *
East Africa		25 (12.0%)	29.7 (18.5–65.6)	64.0	3 (12.0%) [−3.7;9.7]	1.0 *
	Eritrea	16 (7.7%)	32.6 (18.5–65.6)	62.5	3 (18.8%) [−5.3;11.4]	0.473 *
	Somalia	8 (3.8%)	26.2 (24.3–40.9)	62.5	0 (0.0%)	0.603 *
	Other †	1 (0.5%)	24.1 (24.1)	100.0	0 (0.0%)	1.0 *
Eastern Europe		2 (1.0%)	36.4 (34.6–38.1)	50.0	0 (0.0%)	1.0 *
Unknown origin		1 (0.5%)	59.1 (59.1)	100.0	0 (0.0%)	
Total		208		52.9	29 (13.9%) [22.8;35.2]	

* Tested with Fisher’s exact test; # Jordan, Lebanon, State of Palestine; † Ethiopia.

**Table 3 ijerph-20-06331-t003:** Health outcomes according to reported torture exposure status.

Health Outcome	Data Available on Health Outcome (*n*)	Reported Torture Exposure	No Reported Torture Exposure	Chi^2^/Fisher’s Exact Test
Total	208	29	179	
Referral to the Department of Psychiatry	208	16	39	*p* < 0.000
PTSD	208	13	21	*p* < 0.000
Adjustment disorders (ICD-10 F43.2–43.9)	208	0	8	*p* = 0.246
Major mood disorders (ICD10 F30–F39)	208	0	3	*p* = 0.483
Pain conditions *	208	14	95	*p* = 0.631
Hba1c > 44 mmol/mol	171	0	8	*p* = 0.306
Vitamin D deficiency < 25 nmol/L	171	6	56	*p* = 0.653
Latent tuberculosis (Quantiferon test)	207	6	28	*p* = 0.504
TSH < 0.3 and >4.5 × 10^−3^ IU/L	171	2	10	*p* = 0.779
Anemia(Hb female < 7.3. Male < 8.3 mmol/L)	171	0	24	*p* = 0.036
Hepatitis B	167	0	1	*p* = 0.687
Hepatitis C	167	1	2	*p* = 0.329

* Information on self-reported pain conditions was collected through semistructured interview. Only non-transient pain was assessed, and only pain conditions stated by the participant or the physician to negatively affect participation in or quality of social or working life were included.

## Data Availability

Not applicable.

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
