# Peer review of "What Are the Characteristics of Torture Victims in Recently Arrived Refugees? A Cross-Sectional Study of Newly Arrived Refugees in Aarhus, Denmark"

_ijerph, 2023, doi:10.3390/ijerph20146331_

Round 1
Reviewer 1 Report
The authors cover an important subject, i.e. of torture survivors in refugee populations in EU host countries, specifically for refugees arriving in Aarhus between 2017-2019. The methodology is not easy to identify, and might require an easier- to- read and more specific description. If I understand correctly, a database using information on referrals in newly arrived torture survivors in Aarhus was created and analysed, including records from the interviews conducted by doctors (?) and probably others with the refugees.
From this process, the authors tried to confirm, that the UN definition criteria for (alleged !?) torture were fulfilled and if ICD 10 criteria were fulfilled in clinical estimate.
This is partly problematic. I think the appraoch of confirming UN CAT criteria is a very good idea, but it is not clear if this is just speculation based on (probably incomplete) documentation by doctors or lawyers (not clear), or based on a standard instrument, an examination with the UN Istanbul Protocol or on an interview with lawyers.
The same problems arise with PTSD diagnosis (in the text "information from the PTSD clinic regarding diagnosis of PTSD (using ICD-10 standards)"- which is in several ways problematic- did doctors use a standard method to diagnose PTSD ? (such as HTQ, CIDI) and how are they trained ? Clinical notes alone are not up-to-date standard.
Why not other diagnostic categories (MDD has a very high comorbidity in addition to other comorbid problems after torture) or impairment, - the singular focus on Torture sequels is problematic ..
Further, many refugees report on torture experience much later, even if their asylum claim might benefit from reporting earlier.
Finally, this is a culturally heterogenous sample, with a high percentage of Syrians, with no information on cultural factors such as idioms of distress (see the excellent UNHCR hand book on idioms of distress in Syrians by Kirmaier).
If we consider the original populations, torture exposure has different forms, and geographical or ethnic target, so the data focus on a relatively small but still heterogenous and selected sample, which limits conclusions .
The statistics looks ok, but might merit a statistical review. Still, even a good statistical approach is not helpful if the way data are collected is problematic, as in this case, and might create an incorrect impression of methodological rigidity.
Still, the study utilizes very good ideas, such as the use of UN CAT criteria for alleged (sic) torture, and it is good to have data.
This might be seen as a pilot study, to be followed by an improved approach at a later stage. Major conclusions or generalisations are in my opinion not sufficiently evidence based using this data.
Overall, I I think the study should be published after significant adaptation, especiallly discussing the above limitations.
The above
Reviewer 2 Report
Dear Authors,
Thank you for having me review your submission. Although the intended contribution of the article is certainly interesting, a range of technical issues currently hinders the further development of the paper into a publishable article. I am listing the most important issues I have detected:
1. The authors focus on the characteristics of torture victims in the subject, which raises important questions regarding the (eventual) generalizability of the findings. It is currently also unclear if the object of the study is really the characteristics of torture . I suggest to improve the reporting on the research design, and even more importantly, reflect on the research design, and verify if the current design is adequate for answering the research questions.
2. The paper is not grounded in the prevalence of torture and trauma history research. Recent theoretical development and understanding of this sector are not incorporated.
3. Core constructs, such as the prevalence of torture, PTSD, etc. are not specifically and not precisely defined, which makes the discourse rather woolly. All core constructs need to be precisely defined, in line with their use in recent public health.
4. It is important to study recent developments in public health research, and use conceptualisations and theories consistent with their use in that research.
5. It is unclear how the study was designed. Much more information needs to be shared, for the reader to understand how the inner model (theory) is connected to the outer model (measurement)... It is currently also unclear how the core constructs in the study have been operationalized. This probably cannot simply be remedied by providing insight into the design of the measurement instrument (questionnaire), but ecological validity needs to be demonstrated.
I am sorry that I cannot be more positive regarding the article, but I hope my suggestions will help the authors improve their research skills and I wish them good luck with the study.
Reviewer 3 Report
Overall, the paper is interesting and adds to our understanding about the effects of torture on refugees.
I would like to see more information from the data. If these refugees were given a general health assessment, what other information was included that could be useful. You could also do a split file and compare non-tortured refugees with tortured refugees to see if there are any significant differences between these two groups. Are there health ailments that exist in both groups? Are there health ailments common to one group but not the other. I think you have the ability to talk more about not only torture victims, but refugees in general. Even if you just focused the discussion on tortured refugees, that would greatly improve the paper. Knowing more about this population is important and even if you provide a descriptive account, this will add to our overall understanding of refugees, and those who experience torture. At the end of the paper, I was left wanting to know more about the health outcomes of their sample.
Round 2
Reviewer 2 Report
Critique 1: Strengthening the discussion section Based on the provided abstract, it appears that the discussion section of the paper could be significantly strengthened. The authors should elaborate on the implications of their findings, particularly in relation to the prevalence of reported torture and the associations between demographic characteristics, torture exposure, and PTSD. By providing more in-depth analysis, the authors can offer valuable insights for readers and contribute to the existing body of knowledge on this topic.
Critique 2: Recommendations for international organizations and administrators The paper could benefit from including specific recommendations for international organizations, administrators, or custodian institutions involved in the rehabilitation and support of torture survivors. By outlining actionable steps and strategies that these entities can implement, the authors can help translate their research findings into real-world impact.
Critique 3: Gender disparity in torture exposure The abstract highlights a significant difference in reported torture exposure between males (11.5%) and females (2.4%). The authors should discuss this disparity in greater detail, examining possible explanations for the observed gender difference and the potential implications for intervention and support services.
Critique 4: Expanding on the regional origins of refugees The abstract mentions that the largest groups of refugees in the study originated from Syria, Iran, Afghanistan, and Eritrea. The authors should provide further information about these regions, such as the political and social contexts that may contribute to torture exposure. This additional context will help readers understand the broader factors influencing the study's findings.
Critique 5: Limitations of the study The authors should discuss the limitations of their study, including potential biases in self-reporting, sample size, and generalizability of the findings. Acknowledging these limitations will provide a more balanced perspective on the research and help guide future studies in this area.
In conclusion, the paper could be improved by strengthening the discussion section, providing recommendations for international organizations and administrators, discussing the gender disparity in torture exposure, expanding on the regional origins of refugees, and addressing the limitations of the study. The final decision on the manuscript's suitability for publication lies with the journal editor.
Reviewer 3 Report
The changes made are good.
